# A Promising Biocompatible Platform: Lipid-Based and Bio-Inspired Smart Drug Delivery Systems for Cancer Therapy

**DOI:** 10.3390/ijms19123859

**Published:** 2018-12-04

**Authors:** Min Woo Kim, Seung-Hae Kwon, Jung Hoon Choi, Aeju Lee

**Affiliations:** 1International Research Organization for Advanced Science and Technology (IROAST), Kumamoto University, Kumamoto 860-8555, Japan; minwoo-kim@kumamoto-u.ac.jp; 2Division of Bio-imaging, Chuncheon Center, Korea Basic Science Institute (KBSI), Chuncheon 24341, Korea; kwonsh@kbsi.re.kr; 3Department of Anatomy, College of Veterinary Medicine, Kangwon National University, Chuncheon 24341, Korea; jhchoi@kangwon.ac.kr; 4Magnesium Research Center, Kumamoto University, Kumamoto 860-8555, Japan

**Keywords:** lipid-based drug delivery systems, biomimetics, functionalization, controlled release, cancer therapy

## Abstract

Designing new drug delivery systems (DDSs) for safer cancer therapy during pre-clinical and clinical applications still constitutes a considerable challenge, despite advances made in related fields. Lipid-based drug delivery systems (LBDDSs) have emerged as biocompatible candidates that overcome many biological obstacles. In particular, a combination of the merits of lipid carriers and functional polymers has maximized drug delivery efficiency. Functionalization of LBDDSs enables the accumulation of anti-cancer drugs at target destinations, which means they are more effective at controlled drug release in tumor microenvironments (TMEs). This review highlights the various types of ligands used to achieve tumor-specific delivery and discusses the strategies used to achieve the effective release of drugs in TMEs and not into healthy tissues. Moreover, innovative recent designs of LBDDSs are also described. These smart systems offer great potential for more advanced cancer therapies that address the challenges posed in this research area.

## 1. Introduction

Systemic treatment with chemotherapeutics remains the conventional way of treating many cancers, despite the serious damage long-term chemotherapy can cause in healthy tissues [1,2,3,4]. Surgical exclusion, radiation therapy, and combinatorial approaches have also been suggested as treatment options, but these modalities cannot be used to kill malignant cells that have already spread through a body [5]. Although anti-cancer agents with relatively lower side effects have been discovered, most have issues, such as drug resistance, lack of drug solubility, and healthy cell damage at effective doses, which are major hurdles to U.S. Food and Drug Administration (FDA) approval. This situation has led to the development of drug delivery systems (DDSs) that help overcome the limitations of conventional treatment approaches [6]. 

Multidisciplinary developments of efficient DDSs have focused on improving therapeutic efficacy, by taking into consideration several biological barriers and tumor microenvironments (TMEs) [7,8]. Here, we present those sequential biological obstacles that trigger bio-interactions with DDSs in the context of cancer therapy (Figure 1). Despite several obstacles in vivo, the functionalization of DDSs has provided a means of causing the accumulation of chemotherapeutics in the vicinity of tumors [9,10,11]. Well-designed DDSs have the advantages of targeted delivery, controlled release, prolonged blood circulation, and reduced immune stimulation, which hinders the premature release and degradation of drugs.

Many therapeutic strategies have achieved popular practical applications, but DDSs still face challenges associated with safety, and this has led to the development of safer DDSs composed of biocompatible substances [12]. In this respect, lipid-based drug delivery systems (LBDDSs), which consist of a variety of lipid components, have been proposed as safer candidates for cancer therapy. LBDDSs have been extensively studied, and are expected to be applied as biodegradable systems for cancer therapy [13]. There are several different versions of lipid-based carriers that stem from manufacturing methods and the main components used. For example, liposomes, micelles, nanoemulsions, solid lipid nanoparticles, core-shell-type lipid-polymer hybrids, biomimetic vesicles, and even blood cells have been widely investigated for lipid-based drug delivery [14,15,16,17,18,19,20]. Table 1 demonstrates the various types of LBDDSs used for cancer therapy that have achieved remarkable results. These systems provide a variety of benefits, including (a) simple modification for multifunctional applications, (b) sufficient capacity for loading multiple agents with diverse properties simultaneously, (c) flexibility to control the intended size of nanoparticles, and (d) the ability to minimize carrier-related toxicities. LBDDSs have cell membrane-like properties and similar ingredients, and these biological resemblances have facilitated the LBDDS applications [21].

A vast amount of research and development on functionalization strategies is being invested to devise efficient DDSs. Moreover, progress in material sciences has accelerated the adoption of methods to utilize synthetic materials simultaneously in addition to lipidic systems. Consequentially, understanding of recent trends is required to prospect new drug delivery candidates. In this review, we briefly summarize strategies based on LBDDSs and considerations of biological barriers to drug delivery. In addition, we introduce new emerging challenges and describe recent trends aimed at improving the efficiency of lipid-based formulations, and discuss future perspectives of LBDDSs.

## 2. Strategies for Prolonged Blood Circulation

Conventional chemotherapies achieve low accumulations at target sites, and have substantial off-target effects that result in critical side effects [30]. Packaging drugs within carriers can prevent unintentional spreading to healthy tissues [31]. To maximize drug accumulation within tumors, drug carriers that can circulate for longer in the blood are required, because this enhances the efficiency of passive targeting (details are discussed in Section 3), which is also referred to as the enhanced permeability and retention (EPR) effect [32]. The main means used to promote the EPR effect is to control the size of carriers with respect to increasing the probability of reaching their intended destination, by avoiding immune response and maximizing extravasation. Recognition by the immune system and subsequent elimination of drugs from blood is the major cause of the rapid degradation of DDSs [33]. Consequentially, we should continue to attempt to identify more efficient ways of avoiding an immune response and circulatory half-lives, despite the low immunogenicities of LBDDSs.

### 2.1. Consideration of Physicochemical Characteristics

#### 2.1.1. Particle Size

The mononuclear phagocyte system, which is a network of immune cells that plays a key role in the uptake and elimination of administered drug carriers, is largely activated by the physicochemical characteristics of LBDDSs, such as size, shape, hydrophilicity/hydrophobicity, and surface charge. Of these, particle size has the greatest effect on immune responses [34]. Smaller drug carriers basically exhibit better blood circulation and less immune stimulation, whereas extremely small particles with hydrodynamic sizes of <20 nm can be easily cleared by renal filtration [8,35]. Thus, most LBDDSs used for cancer therapy are generally designed to be between 20 to 200 nm to favor the EPR effect.

The encapsulation of drugs in conventional lipid micelles of <60 nm is obviously advantageous in terms of pharmacokinetics and drug bio-distributions, compared with liposomal formulations. Liposomes, which are typical LBDDSs, have vesicle sizes ranging from nanoscale to microscale, and thus do not circulate for as long as micelles; however, they can carry greater drug loadings [36]. In addition, the size and lamellarity of liposomes are easily controlled during their preparations by using different techniques, such as membrane extrusion, solvent evaporation, and ultrasonication [37,38]. Consequentially, the size and type of drug carriers should be decided based on considerations of tumor location, tumor type, physicochemical properties of drugs, and dosages needed for effective treatment.

#### 2.1.2. Particle Shape

Recent studies have suggested that micron-sized particles do not necessarily stimulate immune response more than nanoparticles [39]. In fact, particle shape is also an important determinant for half-life in blood. Different particle shapes, such as needle-like, cubic, discoid, and others, can promote adhesion to vessel walls or circulating immune cells [40,41,42]. The reason why different shapes affect blood circulation is unclear, and the fact that most LBDDSs are a naturally spherical shape is rarely a consideration. However, we need to continually take an interest in the impact of a particular shape or size on blood circulation to promote multidisciplinary research on various hybrid DDSs.

#### 2.1.3. Hydrophobicity

When drug delivery systems enter the bloodstream, they can trigger innate and adaptive immunity, but macrophages cannot immediately recognize drug carriers. In order to make them visible to immune cells, the surfaces of drug carriers should be marked by opsonization, a process whereby specific serum components bind to foreign materials [43,44]. This process is greatly influenced by surface hydrophobicity, which can result in the adsorption of serum proteins onto the surfaces of particles. On the other hand, amphiphilic phospholipids, which are main components of LBDDSs, can hide hydrophobic properties by extending hydrophilic lipid heads outward. This property of LBDDs facilitates in vivo applications with hydrophobic drug formulations [45]. 

Lipid micelles are composed of a hydrophobic core and a hydrophilic shell, and provide refuges for poorly water-soluble drugs and high drug-loading capabilities in biological environments. On the other hand, liposomes have a cell-like structure comprised of an aqueous inner cavity surrounded by an amphiphilic lipid bilayer. Liposomes can encapsulate diverse molecules—for example, water-soluble molecules can be loaded into cores, and hydrophobic molecules can be incorporated into lipid bilayers. Furthermore, this structure allows liposomes to carry large amounts of hydrophilic and hydrophobic drugs in a single vesicle [46]. Importantly, micellar and liposomal structures enable immune surveillance avoidance in blood, the strategic release of their cargoes, and extend circulating half-life [45]. In addition, inhibition of opsonization-induced complement activation can be enhanced by incorporating polyethylene glycol (PEG) into lipid carriers [47,48]. 

#### 2.1.4. Surface Charge

Surface charge plays a key mediatory role in the blood clearance of LBDDSs. For example, liposomal carriers with neutral or negative surface charges can avoid early opsonization [49]. On the other hand, neutral and negative particles may disrupt cellular internalization, because of the repulsion generated between negatively charged cell membranes and particles. Cationic liposomes provide high cellular internalization, but induce massive non-specific binding of serum proteins simultaneously [50]. Hence, it is better for the positively charged lipids to be concealed during circulation, and to reveal them after pH-activated charge reversal in endosomal environment [51,52].

Some studies have shown anionic, neutral, and cationic LBDDSs discharge cargoes via different endocytic pathways after arriving at tumors [53]. These cellular uptake pathways are commonly classified as clathrin-mediated endocytosis, caveolae-mediated endocytosis, micropinocytosis, and receptor-mediated endocytosis. Furthermore, the process adopted in specific cases depends on the type of cancer and on the nature of the ligands attached to the carrier surfaces. Thus, the designs of LBDDSs should be based on understanding the cellular internalization mechanisms.

### 2.2. Polyethylene Glycolylation

The most representative example of a “stealth” lipid-based drug delivery system is Doxil, which was the first liposomal drug approved by the FDA for the treatment of cancer. The liposomes of this doxorubicin formulation are covered by hydrophilic PEG molecules that create a surface shell by extending into aqueous media, which cloaks liposomes from the immune system and prolongs half-lives in the systemic circulation. Presumably because of the commercial success of Doxil, the stealth strategy has led to popular and diverse options, such as poloxamers, polysaccharides, and PEG derivatives with shielding effects similar to PEG, being extensively investigated [54].

Many different types of lipid–PEG derivatives—that is, functional lipids conjugated with various materials, such as targeting molecules, imaging molecules, and other functional polymers—have been demonstrated to act as effective drug carriers. Thus, lipid–PEG derivatives are widely utilized to protect drugs because of their convenience and functionalities [55]. Lipid–PEG derivatives are able to fuse with LBDDSs [48]. For example, polymeric micelles, mainly composed of lipid–PEG conjugates, reinforce in vivo applications, and have also been functionalized to improve pharmacological effects. However, anti-PEG immunological response after repeated injections of PEGylated drug carriers may result in accelerated blood clearance [56], and PEG layers could interfere with drug release and LBDDS internalization [57], which underlies the need for rational drug carrier design [58].

### 2.3. Biomimicry Inspired from Cells

Biomimetic LBDDSs have been produced by modifying the cell membranes of erythrocytes, leukocytes, exosomes, platelets, and cancer cells. In particular, red blood cell (RBC) membranes provide a valuable alternative to the use of PEG [59]. CD47 proteins, which are highly expressed in RBC membranes, emit a “do not eat me” signal, and protect RBCs from phagocytosis by binding to SIRPα (signal regulatory protein alpha), an inhibitory receptor expressed on macrophages. The use of RBC drug carriers offers a means of extending drug circulatory half-lives and inhibiting unintended drug uptake by healthy tissues. Lizano et al. reported that the incorporation of a lipophilic drug into mouse erythrocytes increased circulatory half-life more than 10-fold as compared with PEGylated micellar drug carriers [60]. Furthermore, the use of RBC also reduces drug toxicities compared with free drugs. Recently, Brenner et al. suggested a RBC hitchhiking strategy, whereby drug carriers adsorbed onto RBCs would be used to avoid dominant liver uptake and to improve the deliveries of a wide range of drug carriers [61]. However, the delivery of micron-sized whole RBC platforms (RBC 5–8 μm; RBC ghost 1–2 μm) are adversely affected by insufficient extravasation and instability. 

To address these challenges, some have suggested a way of delivering drugs using nano-sized erythrosomes (liposome-like RBC membrane vesicles) extracted from RBC ghosts [62]. During recent years, lipid–polymer hybrids with a lipid coat and polymeric core have attracted attention as drug carriers [63]. These types of systems possess the characteristics of both lipid carriers and solid polymeric nanoparticles: they are highly stable, and have high loading efficiencies and excellent drug release kinetics. Cell membrane–polymer hybrids have become popular platforms in the wake of the development of the hybrid design [64]. Many biomimetic strategies have been suggested, for more versatile drug delivery vehicles with superior circulation half-lives, including the functionalization of core drugs by coating them with natural erythrocyte membranes.

## 3. Strategies for Targeting the Tumor Region

### 3.1. Passive Targeting

Stable LBDDSs that can stay in circulation longer by avoiding an immune response are able to passively accumulate in tumors. The optimization of long half-life circulation strategies allows drug carriers to selectively penetrate leaky tumor vessels [65]. The EPR effect is the strategy usually used to deliver drugs into the proximities of tumors. The membrane-like structure of LBDDSs offers considerable design flexibility and allows particle sizes to be controlled to maximize EPR. It has been shown that particle sizes of <200 nm exhibit higher levels of extravasation [66,67]. However, Fabienne recently pointed out that thousands of research papers on EPR have only considered TMEs in rodents, and recommended that basic design readjustment of drug delivery systems is necessary to meet human requirements [68]. In particular, human vasculatures are heterogeneous and slowly generated, and thus, clinically drug carriers of <100 nm would be ideal. Furthermore, strategies aimed at controlling factors that mediate blood vessel extension, generation, and permeabilization are helpful [69], and functionalized drug carriers that can release drugs near tumors area would enhance EPR in humans.

### 3.2. Direct Modification with Targeting Ligands

Active targeting involves the recognition of cancer cells, and leads to the maximization of the accumulation and cellular internalization of drugs [70]. Targeting ligands have been attached to the surfaces of LBDDSs by direct coupling [71], or often conjugated with lipid PEG and then simply fused with LBDDSs by the post-insertion method [48]. A variety of active targeting strategies using lipid-based systems have been suggested to elicit the delivery of payloads within tumor tissues. LBDDSs with various ligands have been reported to improve delivery efficiency, as shown by enhanced cytotoxic activity against cancer [72]. Representative advantages and disadvantages of ligands are discussed in Table 2.

Identifying optimal targets so as to enhance the effect of active targeting is critical. The rationale used for selecting appropriate targets involves identifying receptors expressed at higher levels on target cancer cells than on normal cells. Furthermore, TMEs and the expressions of a variety of receptors overexpressed on cancer cells have been reported to be highly correlated with malignant progression, so thus they could be potential therapeutic targets (Figure 2).

#### 3.2.1. Antibodies

Antibodies are well-known to be able to recognize cancers, especially receptors or surface antigens highly expressed on cancer cells. Since the first tumor antigen-targeting monoclonal antibody (mAb) was developed in 1975 [76], a number of mAbs have received FDA approval for cancer therapy; however, some of them are still undergoing clinical trials [77]. In fact, antibody-based immune therapy has been used to treat cancer, and the high affinities and specificities of antibodies for their targets have also been used to guide drug delivery by carriers, including LBDDSs [78,79]. Targeted therapy with mAb-conjugated drug carriers is considered a major potentially curative method, though long-term administration could produce immunological memory against antibodies. Antibody fragments or chimeric antibodies could dramatically reduce immunogenicity against whole antibodies. For example, cetuximab, a recombinant chimeric mAb composed of a murine variable region and human constant region, has been successfully used to treat cancer by targeting the epidermal growth factor receptor [80]. The utilization of dual targeting antibodies is an emerging strategy that improves tumor targeting ability, because antibodies have two epitope binding sites and can react with single or dual targets [81,82].

#### 3.2.2. Aptamers

Aptamers are short, single-stranded DNA or RNA sequences with three-dimensional structures. Systemic evolution of ligands using the exponential enrichment (SELEX) technique led to the synthesis of nucleotide ligands capable of binding to specific targets on cancer cells [83]. Aptamers are considered as more stable ligands than antibodies in vivo, and can be chemically synthesized by in vitro selection, while biological systems are needed to produce antibodies against specific antigens. Synthesis in vitro means that aptamers need to have a wide range of targets. Alshaer et al. summarized attempts made at aptamer-guided drug delivery [84]. PEGylated liposomes or micelles carrying A9, A10, or AS1411 aptamers for targeting prostate-specific membrane antigen and nucleolin, respectively, are examples of aptamer-conjugated LBDDSs [85]. Although aptamers have a short history, and are not still as popular as antibodies, new aptamer-functionalized LBDDSs for targeting cancer are being consistently reported. The low immunogenicity of nucleotide aptamers and lipid carriers make them the most potent combination for in vivo applications, comparable with antibody-guided ones [86]. Zhang et al. suggest that cell-based SELEX has considerable potential, since cancer cells can be specifically targeted without detailed knowledge of the proteins expressed on the cell surfaces; hence, different aptamers can be generated to target different types of cancer [87]. However, aptamers also have limitations, as their affinity is somewhat lower than that of antibodies. Different numbers of epitope binding sites explain the weaker binding affinities of aptamers, which have one binding site. However, multivalent structures constructed using a flexible linker could increase aptamer binding affinity, and other methods of improving aptamer binding affinity have been suggested [88].

#### 3.2.3. Other Ligands

Ligands, such as transferrin, hyaluronic acid, folate, and synthetic peptides with short amino acid sequences, have been suggested as a means of recognizing target cancer cells [78]. Such ligands or small peptides have extremely high specificities for particular target receptors on the surfaces of cancer cells. Transferrin is a serum glycoprotein responsible for transporting iron in blood, and binds to transferrin receptors on cell surfaces, which results in iron internalization. A number of studies have reported that transferrin receptors are upregulated in malignant cancer cells, including those of bladder, brain, breast, and lung cancer, and lymphoma, due to their iron demands [89,90,91]. The development of transferrin-conjugated drug delivery systems allows drugs to infiltrate cells by receptor-mediated endocytosis [92]. Hyaluronic acid (HA) is a molecule with targeting specificity for the CD44 receptor. This linear glycosaminoglycan can bind to the surface of CD44-overexpressing cancers, which include cancers of the head and neck, as well as gastric, colon, prostate, hepatic and breast cancers [93,94]. However, the molecular weight of HA varies, which could affect cellular uptake [78]; thus, the efficacies of drug carriers might depend on the molecular weight of HA, which suggests that HA with an appropriate molecular weight will be selected for specific targets [95,96]. Folate is a water-soluble vitamin B9 that facilitates cellular internalization by interacting with the folate receptor [97]. The benefit of targeting the folate receptor is that its expression in normal tissues is limited, whereas it is highly expressed by several cancers, especially cancers that affect women, such as, cervical, breast, and ovarian cancer [79,98]. These inherent properties of natural ligands facilitate access to tumors and enhance therapeutic effects.

Cell targeting peptides (CTPs) are small peptides chemically synthesized from peptide libraries, and have also been used as targeting ligands [99]. CTPs contain only around 10 amino acids, and are more stable than conventional antibodies [100]. Targets are identified by the amino acid sequences of CTPs, and optimal sequences for interactions with specific cancer cell surface receptors are critical for target recognition. The most extensively studied example of a CTP is the arginylglycylaspartic acid (RGD) peptide [101,102], which has high affinity for integrin receptors overexpressed on the surfaces of several types of cancer cells. In addition, CTP targeting of TMEs using matrix metalloproteinase (MMP) family members is a widely used targeting strategy, although cancer cells are not directly targeted [103,104].

### 3.3. Site-Specific Targeting

Extracellular vesicles (EVs) are membrane-contained nanovesicles secreted by many cell types, and include microvesicles, exosomes, and apoptotic bodies. EVs are intracellular lipid vesicles with sizes typically ranging from 50 to 2000 nm in diameter, and are known to participate in cell-to-cell communications [105]. The use of EVs as drug delivery systems has been recently suggested, due to their functions capable of transferring therapeutic agents and several advantages over liposomes (Table 3). Thus, it is possible that loading them with various therapeutics, such as miRNA, siRNA, or chemical drugs [27], as well as a combined treatment of anti-cancer drugs and oncolytic adenovirus encapsulated in EVs, offers a means of drug delivery for cancer therapy [106].

Recently, Vader et al. have discussed recent studies about in vivo biodistribution of EVs, and their biodistribution profile is similar to that of liposomes [109]. Native EVs generally exhibit highest accumulation in the liver, followed by the spleen and lungs; however, the order could be affected by EV doses, the route of injection, and cell sources [110]. EVs are secreted by most cell types, including fibroblasts, endothelial, epithelial, neuronal, immune, as well as cancer cells [111]. They can act as messengers, and those derived from tumors cells could carry signaling molecules to surrounding cancer cells [112]. In addition, Haney et al. demonstrated that macrophage-derived exosomes highly accumulate in microglial cells and brain neurons [113]. EVs are extracted from cells, thus expressing tumor-targeting ligands on the cell membranes by genetic engineering, which could facilitate cancer-specific drug delivery [114]. 

This approach to using EVs as DDSs has already resulted in considerable success, but the active exploitation of exosomal DDSs has proven to be problematic. The biggest problem is that only small amounts can be isolated from cells, which makes in vivo and clinical investigations difficult [115]. Thus, wide applications of EVs as drug carriers awaits the development of new isolation methods for extracting large quantities. New design-improving performances of EVs are continuously required. For example, Sato et al. and Lin et al. recently suggested the use of exosomes fused with liposomal systems to overcome this availability issue [116,117].

Cancer cells have a variety of characteristics, including infinite reproducibility, immune escape, and homogenous binding properties [15]. These properties are critically important for the spread of tumors from primary lesions to distant locations [118]. Inspired by the characteristics of metastatic cancer, various cancer cell-derived DDSs have been designed to diagnose and treat tumors. The cancer cells could be easily obtained by in vitro culture. Hence, the utilization of natural cancer cell membranes as drug delivery systems offers the promise of the large-scale manufacture of materials capable of encapsulating therapeutics. Furthermore, it has been shown that cancer cell membrane-derived vesicles exhibit low immunogenicity and homologous targeting effects, as well as providing efficient drug delivery [29].

## 4. Strategies for Controlled Release

In this section, we mainly discuss recent advances in stimuli-responsive drug release. Stimuli-responsive LBDSSs are smart drug carriers that can release drugs in response to endogenous (e.g., pH, temperature, redox, etc.) or exogenous (e.g., light, ultrasound, magnetic field, etc.) stimuli.

### 4.1. Typical Drug Release

The developments of methods capable of delivering hydrophobic and hydrophilic drugs have been on-going for some time. Lipid drug carriers must stably capture drugs and controllably release them when they arrive at target sites. For instance, conventional liposomal structure favors the encapsulation of hydrophilic anti-tumor drugs like doxorubicin, because of its ability to traverse the lipid bilayer; however, captured drugs are prone to leak out of liposomes, which cannot be relied upon to retain drugs prior to reaching targeted destinations. Nanoemulsions provide another example of drug-encapsulating LBDDSs. These oil-in-water or inverted phase emulsions are stabilized by surfactants, and thus, could be used to emulsify lipophilic or hydrophilic drugs. However, nanoemulsions quickly release drugs (within 1 min) of administration in vivo [20]. 

As mentioned above, core-shell lipid hybrids—that is, lipid-coating vesicles with a polymer core—have been shown to act as stable drug carriers, and polylactide (PLA), poly-lactide-co-glycolide (PLGA), poly-alkyl cyanoacrylate (PACA), and many other polymers have been utilized in cores. These functional lipid–polymer hybrids exhibit good bioavailability, are stable, and have controllable release characteristics, but the polymer cores are disassembled slowly in vivo (over weeks), and thus, multiple administrations could result in unacceptable accumulations and potential toxicity [119]. Recently, solid lipid nanoparticles (SLNs) were found to have several potential applications. SLNs are 60 to 1000 nm spherical lipid particles with a solid lipid core, and can solubilize hydrophobic drugs. They are produced by emulsifying the lipid drug matrices using surfactants, such as, Tween 20, poloxamer, or polyvinyl alcohol [120]. The biodegradability of lipid components makes them a more attractive proposition than core-shell hybrids. Degradation of lipid cores is more rapid than that of synthetic polymer cores, and results in the production of minimal amounts of toxic metabolites [121]. Moreover, SLNs have relatively stable solid cores, as compared with the liquid cores of nanoemulsion, so they favor controlled long-term drug release. Thus, SLNs combine the merits of nanoemulsion and synthetic polymers, and provide high drug loadings and long-term in vivo stability.

### 4.2. Endogenous Stimuli-Induced Drug Release

Tumor microenvironments are acidic (pH ~6.5), because tumors grow quickly and produce massive amounts of lactic acid. This is a major consideration for drug delivery from the perspective of pH-induced drug release and cellular internalization [122]. Drug carriers sequentially encounter early endosomes (pH ~6.5), late endosomes (pH ~5), and lysosomes (pH 4.6) after being internalized into cancer cells via endocytosis, and must release their cargos promptly before they are fused with lysosomes and removed. A pH-responsive drug release can be achieved by drug carriers that are stable under normal physiological conditions (pH 7.4), but which release their cargo in acidic environments [123]. For instance, the transmembrane gradient method for remote loading of doxorubicin into liposomes has been well-known as a pH-sensitive strategy for liposomal carriers [124]. The solution within liposomes is composed of ammonium salt-based buffers (citrate, phosphate, sulfate, acetate, etc.), which can produce transmembrane gradients. The concentration gradient makes doxorubicin diffuse inside the bilayer, protonate, and become trapped in the liposomes. These pH-sensitive liposomes show rapid drug release characteristics in response to a decrease in external pH. Another approach involves the construction of lipid drug carriers containing or incorporating pH-sensitive lipids, such as lipid-(succinate)-mPEG, N-palmitoyl homocysteine (PHC), and cholesteryl hemisuccinate (CHEMS), which in acidic environments destabilize the lipid membranes of drug carriers and cause rapid drug release [125,126,127].

Mild hyperthermia exhibited by tumor tissues, possibly caused by the glycolytic activity needed for cancer growth, and which can augment external heat stimuli, can be used to cause temperature-sensitive drug carriers to release anti-cancer drugs near tumors [128]. The thermo-responsive strategy used in LBDDS design is based on the gel phase transition temperature (Tm) of lipid components. This phase transition involves a shift from the gel phase to the solution phase when the temperature exceeds the Tm [129]. LBDDSs mainly composed of dipalmitoyl phosphocholine (DPPC; Tm = 41 °C) show good drug retention at physiologic temperature (37 °C), but gradually release their payloads in response to mild temperature elevations. Obviously, lipids with unsuitable Tm values, such as hydrogenated soybean phosphatidylcholine (HSPC; Tm = 55 °C) and 1,2-dimyristoyl-sn-glycero-3-phosphocholine (DMPC; Tm = 25 °C), are unsuitable for this type of controlled drug release [130].

Disulfide-bridged lipids that revert to sulfhydryl forms when they react with reducing agents in TMEs can also be used to target tumors [131]. These disulfide-bonded components can retain robust drug carrier structures until exposed to redox environments. Usually, levels of reducing agents, like glutathione, are remarkably higher in tumor cells than in cells of normal tissues [132], and can break disulfide bridges and cause rapid intracellular drug release. A variety of related studies have demonstrated that the redox-responsive strategy confers high specificity to LBDDS-based cancer therapy [133,134,135].

Enzyme-triggered drug release and tumor targeting have achieved great success in the field of drug delivery. Therapeutic drugs could be directly released from drug carriers through the cleavage of specific enzymes that are over-expressed in TMEs in comparison to normal tissues [136]. For example, Dai et al. have demonstrated that MMP2-sensitive PEG–lipid micellar drug carriers contain PEG–MMP2–lipid polymers, and these enzyme-responsive LBDDSs could significantly improve drug release and therapeutic efficacy [137]. The MMP family, produced by multiple components of TMEs, includes important mediators of cancer invasion, metastasis, and angiogenesis, and thus has been widely employed as a potential target for cancer therapy [138]. MMPs can also play important roles in targeting TMEs and the subsequent uptake of LBDDSs. Zhu et al. and Terada et al have suggested that MMP2-sensitive liposomes contain trans-activator of transcription (TET) peptides or galactosylated cholesterols, which function as targeting ligands [139,140]. They are concealed beneath a long chain of PEG at first, and can be revealed after cleavage of MMP2 peptides at the tumor site, thus increasing tumor-specific uptake of LBDDSs.

### 4.3. Exogenous Stimuli-Induced Drug Release

Recently, several developmental approaches have suggested the use of exogenous stimuli-induced drug release. LBDDSs loaded with anti-cancer agents and imaging molecules have been devised based on this strategy. For example, laser irradiation can be used to produce fluorescence or luminescence images, and to simultaneously induce drug release [141]. Chen et al. developed a near-infrared (NIR) responsive liposomal system for rapid drug release [142]. Lipid carriers containing ammonium bicarbonate and cypate dye were found to immediately generate bubbles when exposed to NIR irradiation, which enhanced lipid bilayer permeability and drug release. In addition, 5-Fluorouracil (5-FU), indocyanine green (ICG), and other NIR dyes have been incorporated into LBDDSs, and laser irradiation could destabilize the lipid layer and facilitate drug release at tumor sites [143]. However, wide applications of these chromophores are limited by considerable toxicities and low imaging quality. Thus, to enhance the performance of NIR-responsive drug release and imaging, Yao et al. suggests lipid drug carriers containing up-conversion nanoparticles (UCNPs) with high quantum yields as a means of achieving NIR-induced controlled doxorubicin release and precise tumor imaging [144]. 

Highly-focused ultrasound stimulation can also be used to trigger the controlled release of therapeutic drugs [145]. Ultrasound waves generate bubbles, cause local heating, and disrupt lipid layers, which enables echogenic lipid drug carriers to be used to induce site-specific drug release and ultrasound images [146,147,148]. In addition, ultrasonic stimulation produces mechanical forces that accelerate drug release and allow tumor imaging [149]. Magnetic systems can also produce heat and increase anti-cancer drug accumulation in tumors [150]. Magnetic Fe_3_O_4_ or Fe_2_O_3_ nanoparticles could be utilized in this manner when embedded in lipid hybrid systems [151,152]. Iron cores and lipidic shells offer excellent biocompatibility and biodegradability. These delivery systems exhibit enhanced drug release and act as MRI contrast agents [153].

## 5. Concluding Remarks and Future Perspectives

Lipid-based drug delivery systems, which offer biocompatibility, biodegradability, and functionality, enable the development of effective systems that have exhibited excellent therapeutic effects in preclinical studies and subsequent clinical trials with cancer. Here, we review several examples of LBDDS platforms with distinctive features, and describe how the performances of LBDDSs are dependent on TMEs. Although previous studies on the treatment of cancer with LBDDSs have reported meaningful results, scientists working today in the drug delivery field need to fully understand the nature of TMEs to design the next generation of LBDDSs. Traditional LBDDSs tend to be limited by incomplete drug release, poor loading capacity, and instability, but recent developments of drug delivery systems are offering more engineering benefits that enhance functionality. Natural and synthetic lipids have biologic and physicochemical characteristics that are advantageous. For example, they have long circulatory half-lives, enable targeted therapy, and allow controlled drug release. Furthermore, merging the advantages of polymers with those of LBDDSs is likely to play an important therapeutic role. It appears obvious that more than one strategy is required to meet the demands of cancer treatment, and that multidisciplinary understanding and sharing information might result in new treatment paradigms.

## Figures and Tables

**Figure 1 ijms-19-03859-f001:**
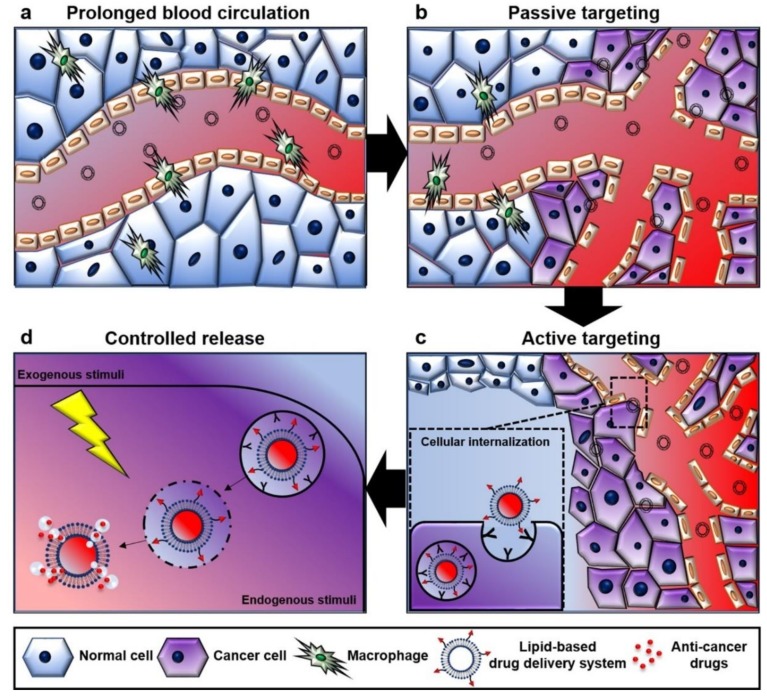
Association of lipid-based drug delivery systems (LBDDSs) with biological systems. Several factors have been considered to increase the delivery efficiency of lipid-based drug delivery systems, including (**a**) prolonged blood circulation, (**b**) passive targeting through the leaky tumor vessels, (**c**) active targeting to penetrate within the tumor, and (**d**) controlled release profile of payloads.

**Figure 2 ijms-19-03859-f002:**
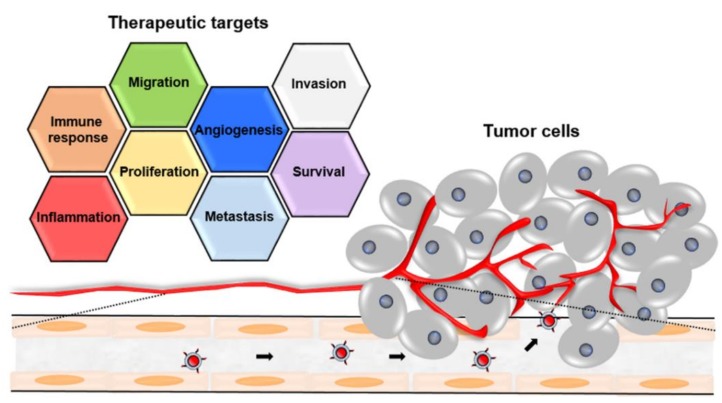
Multiple potential therapeutic targets involved in tumor progression.

**Table 1 ijms-19-03859-t001:** Main lipid-based drug delivery systems and summary of their characteristics.

Type		Core	Lipid lamellarity	Size	Characteristic	Reference
**Micelles**		- Hydrophobic drug	- Monolayer	- 2 nm to 80 nm	- Lipid micelles are small-sized vesicles for solubilization of various poorly soluble pharmaceuticals.	[22]
**Liposomes**		- Hydrophilic - Hydrophobic drug (inner membrane space)	- One to twenty bilayers	- 30 nm to 3000 nm	- Liposomes are synthetically constructed phospholipid vesicles can encapsulate both hydrophobic and hydrophilic drug.	[23]
**Nanoemulsions**		- Hydrophobic drug (O/W) - Hydrophilic drug (W/O)	- Monolayer	- 50 nm to 500 nm	- Nanoemulsions are kinetically stable liquid-in-liquid dispersions with droplet sizes which has high surface area.	[24]
**Solid lipid nanoparticles**		- Solid lipid core-drug matrix	- Monolayer	- 50 nm to 1000 nm	- Solid lipid core instead of liquid oils may provide stability and controlled drug release as the mobility of the drug in a solid lipid matrix.	[25]
**Polymer-lipid hybrids**		- Polymeric core-drug (PLGA, gold, silica, iron oxide and etc.)	- Monolayer - Bilayer	- Polymer core (smaller than typically 300 nm) + bilayer (3 nm to 5 nm)	- Hybrid vesicles have an advanced vesicular structure to integrate best characteristics of liposomes and polymer in a new, single vesicle.	[26]
**Biomimetics**	**Exosomes**	- Hydrophilic/hydrophilic drug	- Bilayer	- 40 nm to 100 nm	- Exosomes are small intracellular membrane-based vesicles with desirable features such as a long circulating half-life, the intrinsic ability to target tissue and biocompatibility	[27]
	**Blood cells (RBC, WBC and platelet)**	- Polymeric core-drug - Hydrophilic/hydrophilic drug	- Monolayer - Bilayer	- 100 nm (nanovesicles) to 8 µm (whole cells)	- Blood cell-based vesicles have many unique advantages such as long life-span in circulation (especially erythrocytes), target release capacities (especially platelets), and natural adhesive properties (leukocytes and platelets).	[28]
	**Cancer cells**	- Polymeric core-drug - Hydrophilic/hydrophilic drug	- Monolayer - Bilayer	- Polymer core (smaller than typically 300 nm) + bilayer (3 nm to 5 nm)	- Cancer cell-derived vesicles carry the full array of cancer cell membrane antigens, and thus offer the inherent homotypic binding phenomenon frequently observed among tumor cells.	[29]

**Table 2 ijms-19-03859-t002:** Advantages and disadvantages of different targeting ligands.

Ligand	Targeted drug delivery		Reference
	Advantages	Disadvantages	
**Antibody**	- Barely affected by nucleases in vivo - Different antibodies to bind to the same antigen which has multiple epitopes - High affinity to the target	- High immunogenicity - Easily loss of activity - Susceptible to high temperature and pH changes - Animal-based production	[73]
**Aptamer**	- Chemical synthesis - Smaller than antibodies (about ten times) - Low immunogenicity - Easy modification	- Susceptible to excess nucleases in a biological system - Low affinity compared to antibody - Limited diversity of possible secondary and tertiary structures	[74]
**Peptide**	- Peptide libraries for almost any desired target - Extremely high receptor affinity - Only around 10 amino acids - High stability	- More expensive and time-consuming - Limited production of peptides with a predetermined length - More potentially active sequences could be missed	[75]

**Table 3 ijms-19-03859-t003:** Comparison of liposomes and extracellular vesicles (EVs).

Vehicle	Advantages	Disadvantages	Reference
**Liposomes**	- Various sizes with either single or multiple lipid bilayer - Diversity of lipid compositions - Mass production - Active clinical research	- Potential complement activation and low cellular uptake - Insufficient drug loading - Premature drug release	[107] [108]
**Extracellular Vesicles**	- Structurally similarity comparable to other membranous structures found in cells - Membrane-associated proteins, receptors, adhesion molecules, as well as a natural corona - Natural characteristics to deliver molecules to target cells - Non-immunogenic	- Limited clinical trials because of uncertainties regarding EVs - Necessity to establish methods for stable production and bulk preparation	

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
