# Peer review of "A Promising Biocompatible Platform: Lipid-Based and Bio-Inspired Smart Drug Delivery Systems for Cancer Therapy"

_ijms, 2018, doi:10.3390/ijms19123859_

Round 1
Reviewer 1 Report
This review paper entitled “A Promising Biocompatible Platform: Lipid-based and Bio-inspired Smart Drug Delivery Systems for Cancer Therapy” summarized many types of functional lipid-based nanoparticles that have been used in tumor treatment. The topic is good, and the structure of this review is well organized. Therefore, I suggest accepting this paper after a minor revision. Some comments and suggestions are listed as below:
1) The first time that the authors mentioned “tumor microenvironment” is in Line 40 (Page1), however, they mentioned the abbreviation of “tumor microenvironment (TME)” at the conclusion section (Page 11, Line 432). I’d like to suggest that the authors should check this kind of errors in the whole manuscript.
2) The authors mentioned “enhanced blood circulation” in Figure 1a and Line 82, which seems a little confusing. I think if “prolonged blood circulation” is better.
3) In “4.2. Endogenous stimuli-responsive drug release” section, I suggest that the authors add a paragraph to mention “enzyme/protease responsive liposomes or lipid-based nanoparticles”, because overexpression of proteases (such as MMPs) is also a very important hallmark of tumor microenvironment.
4) Line 369, the authors mentioned “Tumor microenvironments are acidic (~pH 5.5)”. First, it should be “The tumor microenvironment is acidic”. The pH range of TME is 5.5-7.0 according to most of current reports. Therefore, I think the author used the lower limit “~pH 5.5” is not proper in this sentence, and maybe “~6.5” is better.
5) If the authors can highlight some excellent work by using some figures, it will be better.
Author Response
Responses to reviewers’ comments November 29, 2018
On behalf of all authors, I appreciate a reviewer’s thoughtful comments made for our manuscript (ID: ijms-397758). We amended the manuscript as the reviewer suggested.
- Abbreviations in the whole manuscript were rewritten.
- “enhanced blood circulation” was replaced with “prolonged blood circulation”.
- A further discussion regarding enzyme/protease responsive drug release strategy was added in line 405-417.
- As suggested, “~pH 5.5” was replaced with “~6.5”.
Reviewer 2 Report
Efficient drug delivery has a crucial role in a successful diseases treatment and it is still a challenge in medicine. Lee and coauthors wrote an interesting and review about lipid-based drug delivery systems as strategy for cancer therapies.
However few comments should be addressed:
1) In chapter 3.3 authors should also mention that cancer derived extracellular vesicles have been proposed as delivery vehicles for chemotherapeutic agents and viruses. This should be at least discussed as viruses can act to both primary and metastatic lesions.
2) The biodistribution of EVs has been studied using in vivo imaging technology, this should be at least mentioned as this is of importance for cancer drug delivery
3) A table with advantages and disadvantages about the use of EVs versus lipid-based drug delivery systems could be helpful for readers
4) Authors report that cancer cells, in contrast with exosomes can be easily used as drug delivery systems. This sentence should be reformulated as also EVs can be easily isolated from cancer cells
Author Response
Responses to reviewers’ comments November 29, 2018
On behalf of all authors, I appreciate a reviewer’s thoughtful comments made for our manuscript (ID: ijms-397758). We amended the manuscript as the reviewer suggested.
- A further explanation for extracellular vesicles was added in chapter 3.3
- “Table 3” explaining about advantages of EVs was added.
- The unclear sentence “in contrast with exosomes” in line 335 was removed.